# S/O/W Emulsion with CAPE Ameliorates DSS-Induced Colitis by Regulating NF-κB Pathway, Gut Microbiota and Fecal Metabolome in C57BL/6 Mice

**DOI:** 10.3390/nu16081145

**Published:** 2024-04-12

**Authors:** Xuelin Wei, Juan Dai, Ruijia Liu, Guochao Wan, Shiyu Gu, Yuwei Du, Xinyue Yang, Lijun Wang, Yukun Huang, Pengfei Chen, Xianggui Chen, Xiao Yang, Qin Wang

**Affiliations:** 1School of Food and Bioengineering, Xihua University, Chengdu 610039, China; 23010080013@pop.zjgsu.edu.cn (X.W.); liuruijia@stu.xhu.edu.cn (R.L.); wanguochao@stu.xhu.edu.cn (G.W.); siyu@stu.xhu.edu.cn (S.G.); duyuwei@stud.uni-obuda.hu (Y.D.); yangxinyue2@stu.xhu.edu.cn (X.Y.); wanglijun@mail.xhu.edu.cn (L.W.); huangyk@mail.xhu.edu.cn (Y.H.); 1220180037@mail.xhu.edu.cn (P.C.); chen_xianggui@mail.xhu.edu.cn (X.C.); 2School of Laboratory Medicine, Chengdu Medical College, Chengdu 610500, China; daijuan@cmc.edu.cn; 3Chongqing Key Laboratory of Specialty Food Co-Built by Sichuan and Chongqing, Chengdu 610039, China; 4Department of Nutrition and Food Science, University of Maryland, College Park, MD 20742, USA

**Keywords:** CPAE, S/O/W emulsion, colitis, gut microbiota, fecal metabolism

## Abstract

Inflammatory bowel disease (IBD) has attracted much attention worldwide due to its prevalence. In this study, the effect of a solid-in-oil-in-water (S/O/W) emulsion with Caffeic acid phenethyl ester (CAPE, a polyphenolic active ingredient in propolis) on dextran sulfate sodium (DSS)-induced colitis in C57BL/6 mice was evaluated. The results showed that CAPE-emulsion could significantly alleviate DSS-induced colitis through its effects on colon length, reduction in the disease activity index (DAI), and colon histopathology. The results of ELISA and Western blot analysis showed that CAPE-emulsion can down-regulate the excessive inflammatory cytokines in colon tissue and inhibit the expression of p65 in the NF-κB pathway. Furthermore, CAPE-emulsion promoted short-chain fatty acids production in DSS-induced colitis mice. High-throughput sequencing results revealed that CAPE-emulsion regulates the imbalance of gut microbiota by enhancing diversity, restoring the abundance of beneficial bacteria (such as *Odoribacter*), and suppressing the abundance of harmful bacteria (such as *Afipia*, *Sphingomonas*). The results of fecal metabolome showed that CAPE-emulsion restored the DSS-induced metabolic disorder by affecting metabolic pathways related to inflammation and cholesterol metabolism. These research results provide a scientific basis for the use of CPAE-emulsions for the development of functional foods for treating IBD.

## 1. Introduction

Both Crohn’s disease (CD) and ulcerative colitis (UC) are prevalent types of inflammatory bowel disease (IBD), which is defined by a chronic ulceration of the intestinal mucosa [1,2]. The incidence of IBD (especially Crohn’s disease) has increased significantly worldwide in recent years, and the pathogenesis of this disease is still unclear and may be related to factors such as genetics, the environment, dietary structure, the intestinal mucosal barrier, autoimmune abnormalities, and intestinal microbial changes [3,4,5].

Currently, the treatment for IBD involves the use of medications such as corticosteroids, salazosulfapyridine, and immunosuppressive agents, etc. [6]. However, there is a need to revisit these therapeutic strategies due to their limited effectiveness and harsh side effects. Recent studies have shown that dietary polysaccharides [7], phytochromes [8], and flavonoids [9] can attenuate intestinal colitis by altering the intestinal microbiota composition and increasing the production of short-chain fatty acids [10]. Therefore, the modulation of the host’s gut microbiota has become a crucial objective in the mitigation and management of IBD [11]. Previous research has found that resveratrol can show a potent probiotic effect by increasing the abundance of *bifidobacteria* [12], and dietary cellulite can also improve DSS-induced colitis by manipulating the gut microbiota to suppress the inflammatory reaction [13].

Caffeic acid phenethyl ester (CAPE) is a polyphenol derived from propolis, which has various functions such as scavenging free radicals, anti-inflammatory, improving the colorectal microenvironment, inhibiting colorectal cancer cell differentiation [14,15], inhibiting the overexpression of pro-inflammatory cytokines, and down-regulating the NF-κB signaling pathway [16]. Previous studies have also demonstrated that CAPE can alleviate DSS-induced colitis via intraperitoneal injection [17]. However, the poor oral bioavailability of CAPE greatly limits its therapeutic modalities and application in functional foods [18]. Recent studies have found that oil-in-water (O/W) microemulsion as a delivery vehicle of CAPE can significantly improve the oral bioavailability and bioactivity of CAPE against cancer cells. In a previous study, we also constructed a solid-in-oil-in-water (S/O/W) multilayer emulsion loaded with CAPE for the targeted delivery and release in the colon [19]. However, the direct effects of CAPE on the gut microbiota and the location of colonic inflammation are rare in in vivo experiments.

To understand the role of this CAPE-emulsion with targeted delivery function for the treatment of IBD and the possible underlying mechanisms involved, the DSS-induced colitis mouse model was gavaged using the S/O/W emulsion with CAPE. Hematoxylin-eosin (H&E) staining was used to determine the histopathologic changes in the colon. Hematology, inflammatory cytokines, the NF-κB signaling pathway, and short-chain fatty acids were detected. UPLC-Q/TOF was used to investigate the fecal metabolome, while 16S rRNA high-throughput sequencing was used to identify changes in the gut microbiota. The theoretical groundwork for creating functional meals to treat IBD is established by this study.

## 2. Materials and Methods

### 2.1. Reagents

CAPE (CAS 115610-29-2, purity > 98%) and DSS (CAS 9011-18-1, MW 36,000–50,000 Da) were obtained from Sigma-Aldrich (Shanghai, China). The ELISA kits for detecting interleukin-1β (IL-1β), interleukin-6 (IL-6), interleukin-10 (IL-10), and the transforming growth factor-β (TGF-β) were available from Tiangen Biotech (Shanghai, China). The rabbit antibodies for β-actin (number: AC026, control for WB) were obtained from Abclonal (Wuhan, China). The antibodies of IκB-α (number: 10268-1-AP) were obtained from proteintech (Wuhan, China). The antibodies of P65 (number: BS-0465R) were obtained from Bioss (Boston, MA, USA). HRP-conjugated goat anti-mouse antibodies were obtained from Thermo Fisher Scientific (Cleveland, OH, USA).

### 2.2. Preparation of S/O/W Emulsion with CAPE

The CAPE-emulsion was prepared according to the previous method [19]. Briefly, five milliliters of 1.8 mg/mL of CAPE solution was mixed with 10 mL of a solution containing 2% casein and agitated for a duration of 4 h, then 10 mL of a solution containing 1.25 mg/mL of NaAlg was mixed and heated at 80 °C for 40 min. The mixture mixed with 20 mL of a solution containing 1 mol/L of CaCl_2_ and was treated at 20,000 rpm for 10 min using an ULTRA-TURRAX T25 homogenizer (IKA, Staufen, Germany). The product was homogenized twice at 16,000 psi in a NanoGenizer 30K high pressure microfluidizer (Genizer, Irvine, CA, USA), and then freeze-dried to prepare the nanoparticles. The obtained nanoparticles were dispersed into coconut oil and placed in a refrigerator at 4 °C for 10 min to form S/O liposomes. After mixing 1 g of S/O liposomes with 7 mL of a solution containing 0.2% sodium caseinate and 0.3 g of lecithin, and homogenizing at 20,000 rpm for 2 min by an ULTRA-TURRAX T18 homogenizer (IKA, Germany), the S/O/W emulsion with CAPE was obtained by homogenizing with a NanoGenizer 30K high pressure microfluidizer (Genizer, USA) at 25,000 psi for 10 min.

### 2.3. Animal Experiment

Male C57BL6/J mice (6 weeks old, 20 ± 2 g) were purchased from Chengdu Dashuo Experimental Animal Co., Ltd. (Chengdu, China). The mice were kept in a controlled environment at 25 ± 2 °C with a relative humidity of 65 ± 10% and a 12 h light/dark cycle, and were acclimatized for 1 week with free access to feed and water. Then, the mice were randomly divided into five groups (*n* = 10/group). The mice received a standard diet and normal drinking water as the control group (Control); the mice were fed a standard diet and drinking water containing 3% DSS (*w*/*v*) as the DSS group (DSS); and the mice were gavaged with S/O/W emulsion containing 10 and 45 mg/kg·BW of CAPE and fed drinking water containing 3% DSS as the CAPE treatment groups (LD and HD), respectively. The mice in the Control and DSS groups were gavaged with an equal volume of saline. The experimental procedures are shown in Figure 1A. During the DSS treatment period (day 15 to 21), weight loss, the consistency of the feces, and the composite disease activity index (DAI) scores for rectal bleeding were recorded every day in each mouse. At the end of the experiment (day 22), the mice were sacrificed by CO_2_ euthanasia. The cardiac artery blood, colon, liver, and spleen organs of the mouse were collected and stored at −80 °C.

Every experiment was carried out in compliance with established procedures and approved by the Chengdu Medical College Experimental Animal Ethics Committee.

### 2.4. Disease Activity Index (DAI)

The disease activity index (DAI) was assessed based on the scores of body weight loss, stool consistency, and rectal bleeding, which were recorded each day as previously described [20].

### 2.5. Histopathology of Colitis

Colon length was measured after the mice were dissected [21]. The fixed colon segments were embedded in paraffin after being dried with ethanol and isopropanol for histological examination. After that, the specimens were divided into slides and stained with H&E. Histological grading was assessed according to the inflammatory criteria as the method [20].

### 2.6. Analysis of Inflammatory Cytokines in Colonic Tissue

The inflammatory cytokines in the colonic tissue were detected according to previously reported methods [20] with slight modifications. The colon tissues were weighed and homogenized in 0.5% PBS solution at a ratio of 1:10 (*w*/*v*), then centrifugated (10,000× *g* for 30 min at 4 °C). The cytokines (IL-1β, IL-6, IL-10, and TGF-β) were detected by ELISA kits (Tiangen Biotech, Shanghai, China) according to the standard operating manual.

### 2.7. Western Blot Analysis

Western blot analysis was performed as the previously reported method [22] with slight modifications. The total proteins of colon tissue were extracted with RIPA buffer (the mass ratio of sample:RIPA = 1:10) and supplemented with 1× protease inhibitor cocktail (Roche, Basel, Switzerland) on an ice bath. Then, the supernatant was collected after centrifugation (10,000× *g* for 10 min at 4 °C). The concentration of the protein was measured using the BCA kit (Tiangen Biotech, Shanghai, China). The protein samples of the same amount (loading amount of 40 µg) were separated on a 12% SDS-PAGE gel and transferred onto a PVDF membrane (Sigma Aldrich, Shanghai, China). After being blocked with TBST (tris-buffered saline containing 0.1% Tween 20) containing 5% skimmed milk, the membrane was incubated overnight at 4 °C with primary antibodies (NF-κB p65, 1:2000; IκB-α, 1:2000; β-actin, dilution 1:100,000), and incubated with secondary antibodies (dilution 1:5000) for 2 h at room temperature after washing. Protein bands were detected using the ECL Plus™ Western Blot Detection System (Pierce, Rockford, IL, USA) and imaged with a GIS V2.0 imaging system (Tannon, Shanghai, China).

### 2.8. Gut Microbiota Analysis

Microbial DNA was extracted from mouse fecal samples using a fecal gene extraction kit (Omega Biotek, Norcross, GA, USA). The V3-V4 region of the bacterial 16S ribosomal RNA gene was amplified with primers F 5′-CCTAYGGGRBGCASCAG-3′ and R 5′-GGACTACNNGGGTATCTAAT-3′ by a GeneAmp PCR 9700 thermal cycler (Applied Biosystems, Foster City, CA, USA). The PCR products used for sequencing were obtained from Majorbio Co. Ltd. (Shanghai, China) on an Illumina MiSeq platform (Illumina, San Diego, CA, USA). The raw data were analyzed using Quantitative Insights Into Microbial Ecology (QIIME, version 2.0). The sequences were clustered into operational taxonomic units (OTUs) with a similarity of at least 97% using UPARSE (version 7.1).

### 2.9. Quantification of Short-Chain Fatty Acids (SCFAs)

SCFAs analysis was performed as the previously reported methods [13] with slight modifications. The dried contents of the cecum (50 mg) obtained from each mouse were combined with 500 μL of a solution containing 0.5% phosphoric acid. The mixture was vigorously stirred using a vortex mixer and subjected to ultrasound for 5 min, then centrifuged at 3000× *g* for 10 min at 4 °C to separate the components. The resulting supernatant was then combined with 500 μL of ethyl acetate and mixed for 5 min before being centrifuged again at 8000× *g* for 10 min at 4 °C. The organic phase was filtered for detection using a Nexis GC-2030 gas chromatograph (Shimadzu, Tokyo, Japan).

### 2.10. Fecal Metabolite Analysis

Metabolomic analysis was performed as the previously reported method [23] with slight modifications. Briefly, 50 mg of fecal samples were sequentially mixed with 200 μL of ultrapure water, methanol, and acetonitrile for extracting the metabolites by stirring for 20 s, centrifugating at 10,000× *g* for 15 min at 4 °C, and collecting the supernatant. Then, the supernatants from three solvent extracts were subsequently combined and filtered with a 0.2 μm filter. An equal volume of extract from each sample was mixed as quality control (QC) samples. Fecal metabolites were separated on an Acquity UPLC HSS T3 C18 column (100 mm × 2.1 mm, 1.8 μm, Waters, CA, USA) using an LC30 UPLC system (Shimadzu, Japan) and detected using an X500R high resolution mass spectrometry (AB Sciex, Framingham, MA, USA) with ESI^+^ mode. The data were analyzed using MetaboAnalyst 6.0 (www.metaboanalyst.ca, accessed on 15 March 2022). Pareto scaling was used to eliminate chemical noise in the data, followed by performing multivariate statistical analysis using partial least squares discriminant analysis (PLS-DA). The potential biomarkers were selected based on their variable importance in the project (VIP > 1.0) value and were identified using the HMDB (www.hmdb.ca, accessed on 15 March 2022) and Metlin (metlin.scripps.edu, accessed on 15 March 2022) database. The metabolic pathways involved in biomarkers were annotated according to the KEGG database (www.kegg.com, accessed on 15 March 2022).

### 2.11. Statistical Analysis

The mean ± standard error results were obtained through repeated samples. Statistical analysis involved using SPSS 26.0 (IBM, Chicago, IL, USA) and Origin 2021b (OriginLab, Northampton, MA, USA), followed by Duncan’s shortest significant range test for multiple mean comparisons with analysis of variance (ANOVA).

## 3. Results

### 3.1. The CAPE-Emulsion Alleviated Symptoms of DSS-Induced Colitis in Mice

DAI scores (clinical parameters reflecting the severity of IBD) are used to assess the DSS-induced inflammatory states of the colon, including the parameters for fecal blood, fecal consistency, and weight loss. CAPE-emulsions dose-dependently reduced DAI scores in DSS-induced colitis mice, as demonstrated in Figure 1B. Colon shortening is a morphological indicator of colonic inflammation. As shown in Figure 1C,D, the high doses of CAPE-emulsion treatment significantly reversed colonic shortening in DSS-induced colitis mice. H&E staining reveals the histological damage of colitis in mice with colitis (Figure 2), where the normal features of a typical crypt with many goblet cells in the control group are not as severe as those caused by DSS, resulting in goblet cell failure, crypt distortion, and intense inflammatory cell infiltration. However, the CAPE-emulsion reduced the infiltration of inflammatory cells, enhanced epithelial structure, reorganized goblet cells, and restored DSS-induced colitis-related morphological changes in mice. These results suggest that the CAPE-emulsion can significantly alleviate DSS-induced colitis symptoms. However, there were no significant differences in the body weight and organ indexes between all groups in the experimental period (Appendix A).

### 3.2. The CAPE-Emulsion Reduced the Overproduction of Inflammatory Cytokines by Inhabiting NF-κB Signal Path in DSS-Induced Colitis Mice

Cytokines are important mediators in the pathogenesis of colitis and have the ability to modulate the inflammation of the mucosal immune system [24,25]. In order to evaluate the effect of the CAPE-emulsion on inflammatory cytokines, the concentrations of inflammatory cytokines (IL-1β, TGF-β, IL-10, and IL-6) and concentrations in colonic tissue were analyzed using ELISA. As shown in Figure 3, the CAPE-emulsion can significantly reduce the DSS-induced increase in inflammatory cytokines. Among these cytokines, IL-6 and IL-1β are pro-inflammatory cytokines that can sensitively characterize the degree of inflammation. However, even treatment with low doses of the CAPE-emulsion can significantly reduce the levels of IL-6 and IL-1β in colonic tissue to even lower than those in the control group. IL-10 and TGF-β, which are anti-inflammatory factors, can inhibit the expression of pro-inflammatory factors, thereby preventing excessive inflammatory responses and promoting immune balance [26,27,28]. Interestingly, the CAPE-emulsion also significantly reduced the DSS-induced increase in the expression of anti-inflammatory cytokines. These results indicated that the CAPE-emulsion could not only directly down-regulate the levels of pro-inflammatory cytokines in inflammatory sites, but also maintain immune homeostasis by reducing the levels of inflammatory regulators in the DSS-induced colitis mice.

The NF-κB pathway is thought to be a signaling pathway closely related to the regulation of pro-inflammatory cytokine expression in innate immunity. Pro-inflammatory factors (such as IL-1β) amplify inflammatory signals by activating the NF-κB signaling pathway [29]. As shown in Figure 3E,F, there was no significant difference in the expression of IκB-α between the groups in the experiment. Compared with that in the control group, the protein expression of P65 in the DSS-treated group increased significantly. However, intervention with high doses of the CAPE-emulsion significantly down-regulated the increase in the protein expression of P65 induced by DSS. The NF-κB signaling pathway relies on the p65:p50 dimer to enter the nucleus and activates the expression of target genes [30]. These results suggest that a high dose of CAPE-emulsion treatment can down-regulate the expression of the P65 protein in the NF-κB signaling pathway.

### 3.3. The CAPE-Emulsion Recovered the SCFAs in DSS-Induced Colitis Mice

SCFAs are produced by the anaerobic fermentation of dietary fiber in the intestine and improve the intestinal barrier and reduce intestinal inflammation [31]. SCFAs in the gut include acetate acid (C2), propionate acid (C3), and butyric acid (C4), among others [32]. As shown in Figure 4, the CAPE-emulsion intervention can restore the levels of propionic acid and butyric acid in the gut and significantly increase the content of acetic acid in the DSS-induced colitis mice. The gut microbiota is closely associated with changes in SCFAs levels. These results also implied a possible effect of the CAPE-emulsion intervention on the gut microbiota.

### 3.4. The CAPE-Emulsion Regulated the Gut Microbiota in DSS-Induced Colitis Mice

To understand the effect of the CAPE-emulsion intervention on the gut microbiota in DSS-induced colitis mice, the composition and abundance of the gut microbiota were analyzed via high-throughput sequencing of 16S rDNA V3-V4. With an average sequence length of 419.97 bp, 684,639 pyrophosphate sequences were detected from 12 samples and clustered into 2094 OTUs with 97% similarity using QIIME (version 2.0) software. Alpha diversity analysis of the gut microbiota revealed that the community diversity (Sob, Ace, and Chao1 indices) was significantly reduced in DSS-induced colitis mice, while the CAPE-emulsion intervention could restore the community diversity. However, there was no significant difference in the community diversity (Shannon index, Simpson index) of the gut microbiota between the groups (Table 1).

The 534 species, 411 genera, 219 families, 127 orders, 53 classes, and 19 phyla were identified from the 2094 OTUs. At the phylum level (Figure 5A), *Firmicutes*, *Bacteroidetes*, and *Proteobacteria* were the major phyla in the gut microbiota, and accounted for more than 90% of the gut microbiota. Notably, the relative abundance of *Bacteroides* was significantly reduced from 79.60% to 56.42% by DSS induction, while the CAPE-emulsion intervention significantly reduced the increase in the abundance of *Proteobacteria* in DSS-induced colitis mice. At the family level (Figure 5B), *Muribaculaceae*, *Lachnospiraceae*, *Prevotellaceae*, *Erysipelotrichaceae*, *Rikenellaceae*, and *Oscillospiraceae* were the major families. Abnormal increases in the abundance of *Muribaculaceae* and *Prevotellaceae* were found in DSS-induced colitis mice, but the CAPE-emulsion intervention significantly inhibited this abnormal proliferation. A hierarchical cluster analysis and heatmap of the 55 most abundant genera are shown in Figure 5C. The results showed that the control group was significantly different from the DSS-induced colitis mice, while the high-dose CAPE-emulsion intervention group was farther away from the other groups in the DSS-induced colitis mice. The beta-diversity characterized by principal coordinate analysis (PCoA) revealed differences in the structure of the gut microbiota between the different groups. As shown in Figure 5D, duplicate samples in each group could form aggregates, while each group could be separated from the other group. These findings suggest that the CAPE-emulsion intervention in DSS-induced colitis mice has a significant effect on the gut microbiota.

Linear discriminant analysis effect size (LEfSe) also indicated that the CAPE-emulsion intervention could restructure the diversity of the gut microbiome in DSS-induced colitis mice (Figure 5E–G). In DSS-induced colitis mice, the abundance of *Vibrionimonas*, *Paraprevotella*, *Burkholderia-Caballeronia-Paraburkholderia*, *Sphingomonas*, *Afipia*, *Ideonella*, *Dubosiella*, *Faecalibaculum*, and other bacteria was significantly increased, and the abundance of *Desulfovibrio*, *Acinetobacter*, *Odoribacter*, *Rikenella*, *Alloprevotella*, and other bacteria was significantly reduced, while intervention with the CAPE-emulsion can restore these abundance changes (Appendix A). These gut microbiotas with significant changes in abundance were closely associated with the synthesis of SCFAs and the development of inflammation. Thus, these results also suggest that the intervention of the CAPE-emulsion could inhibit the development of colitis by partially altering the structure of the gut microbiota, in particular by lowering the quantity of pathogenic bacteria and raising the quantity of beneficial bacteria.

### 3.5. Effect of the CAPE-Emulsion on Fecal Metabolome

To understand the effects of the CAPE-emulsion on fecal metabolites in DSS-induced colitis mice, a total of 425 metabolites were found by UPLC-Q-TOF. The PLS-DA model was established using the metabolite abundance of all samples to distinguish differences in the fecal metabolome among the different groups. The cross-validation based on this PLS-DA model indicated that the model was not over-fitted and had good predictive ability (Q^2^ = 0.65, R^2^ = 0.90). As shown in Figure 6A, there were clear differences among the control, DSS, and CAPE-emulsion intervention groups. The metabolite clusters in the CAPE-emulsion intervention group were more similar to those in the control group than in the DSS group, indicating that the fecal metabolome may be regulated by the CAPE-emulsion.

A total of 24 metabolites were identified from the differentially abundant metabolites (VIP > 1) by theoretically accurate mass and secondary mass spectra in the metabolomics databases. The changes in the abundance of these key differential metabolites are shown in Table 2. In the DSS-induced colitis mice, the abundance of 4 differential metabolites was elevated and the abundance of 16 differential metabolites was down-regulated in comparison to the control group. After the CAPE-emulsion intervention, the trend of six differential metabolites was reversed. Metabolite enrichment analysis based on databases (HMDB and KEGG) suggests that these key differential metabolic markers are involved in the steroid hormone biosynthesis (C00468, C00951, C00280, C04295, C18042), steroid biosynthesis (C01673, C15777, C01789), sphingolipid metabolism (C00319, C00836), linoleic acid metabolism (C01595), taurine and hypotaurine metabolism (C05844), tryptophan metabolism (C00780, C05639), tyrosine metabolism (C00780, C05639), and primary bile acid biosynthesis (C01301, C00695). The changes in these biomarkers illustrated the potential biological effects of the CAPE-emulsion intervention in the DSS-induced colitis mice.

To investigate the relationship between fecal metabolites and the gut microbiota, the correlation of key fecal metabolites and gut microbiota was calculated via Spearman correlation analysis (Figure 6B). The results showed that the abundances of gut microbiota constituents, such as *Dubosiella*, *Faecalibaculum*, *Paraprevotella*, and *Bifidobacterium*, which increased under DSS induction, were negatively correlated with the metabolic markers involved in steroid biosynthesis, sphingolipid metabolism, and primary bile acid biosynthesis. The intestinal flora whose abundance was down-regulated under DSS induction, such as *Marvinbryantia*, was positively correlated with the metabolic markers involved in steroid biosynthesis, sphingolipid metabolism, and primary bile acid biosynthesis.

## 4. Discussion

Recently, with the increasing prevalence of IBD, more researchers have been searching for approaches to prevent and alleviate IBD (UC and CD). CAPE has attracted much attention due to its multiple functions in improving human intestinal health [17]. In this study, an S/O/W emulsion with a targeted release of CAPE in the colorectum was used to treat mice with DSS-induced colitis. The results showed that this CAPE-emulsion can effectively alleviate colitis symptoms as determined by DAI, colon length, and colon pathology.

Some studies have shown that T-helper (Th) cell subsets are directly related to the impact of autoimmunity and autoimmune inflammatory diseases in IBD [2]. Damage to epithelial cells by enteric pathogens or irritating drugs (e.g., DSS) promotes the expansion of Th 1/Th 2 cells. The proinflammatory cytokines secreted by these immune cells drive an acute inflammatory response to reject infection [2]. To control excessive inflammatory responses, gut microbiota promotes the expansion of regulatory T (Treg) cells, and the IL-10 and TGF-β produced by Treg cells will drive the reduction in pro-inflammatory cytokines, leading to Th cells apoptosis and the regression of mucosal lesions [3]. In this study, the levels of pro-inflammatory cytokines and anti-inflammatory cytokines secreted by the colon in the CAPE-emulsion treatment groups were significantly lower than those in the DSS group and control group, which implies that in the long term, the CAPE-emulsion intake can not only effectively suppress inflammation by reducing the secretion of pro-inflammatory cytokines, but also control the degree of inflammation and shorten the process of an inflammatory response.

Many studies have proven that the NF-κB signaling pathway is crucial for the inflammatory response and can regulate the expression of a variety of pro-inflammatory cytokines [33]. When inflammatory cytokines, growth factors, or chemokines in the colon activate the NF-κB signaling, IκBα will be phosphorylated and degraded, and the dimer of P65:P50 is transferred to the nucleus to cause an inflammatory response [29]. In this study, the CAPE-emulsion treatment could significantly reduce the expression of P65 in colon tissue, implying an effect of the CAPE-emulsion treatment on the NF-κB signaling pathway.

SCFAs (mainly acetate, propionate, butyrat) are produced by the anaerobic fermentation of dietary fibers in the gut. Some studies have shown that SCFAs produced by gut microbiota improve intestinal barrier function and reduce intestinal inflammation. For example, butyrate is an important energy source for colon cells and can promote colon cancer cell apoptosis and reduce intestinal inflammation [34], and an impaired supply of butyrate may be related to decreased gastrointestinal function [35]. Propionic acid can induce T-cell differentiation to inhibit intestinal inflammation [36]. Acetate activates G protein-coupled receptors and inhibits histone deacetylase activity, thereby potentially preventing metabolic and inflammatory diseases, and playing a key role in the inhibition of enteropathogens by *bifidobacteria* [35]. In this study, the CAPE-emulsion promoted the production of SCFAs, especially butyric acid, which suggested that affecting SCFAs is also one of the mechanisms by which the CAPE-emulsion inhibits DSS-induced colon inflammation.

In past studies, gut microbiota has received increasing attention as a potential driver of the inflammatory process in IBD patients. Several studies have found that the gut microbiota imbalance in IBD patients, with a loss of overall diversity, a decrease in *Firmicutes*, and an increase in *Proteobacteria* [37,38], is similar to the results of this study. Members of the phylum *Proteobacteria* are less abundant in the intestines of healthy people, and gut microbiota imbalance is often caused by a sustained increase in the abundance of *Proteobacteria*. Therefore, an increased abundance of *Proteobacteria* has been used as a potential diagnostic criterion for gut microbiota imbalance and related diseases [39]. Among those genera whose abundance is down-regulated by the CAPE-emulsion, *Burkholderia-Caballeronia-Paraburkholderia*, *Vibrionimonas*, *Afipia*, *Sphingomonas*, and *Ideonella* belong to the *Proteobacteria* phylum, which is the characteristic bacteria involved in IBD-related inflammation [40]. These microorganisms can adhere to and invade intestinal epithelial cells, destroy the integrity of the intestinal epithelium, and transform the intestine into a pro-inflammatory niche suitable for the proliferation of pathogenic bacteria, leading to gut microbiota dysbiosis [41,42]. Among the bacterial genera whose abundance was restored by the CAPE-emulsion, *Odoribacter* had confirmed to promote the growth of SCFAs [43]. Studies have shown that *Odoribacter* effectively alleviates colitis in mice through a mechanism that may be related to the regulation of neutrophil extracellular trap formation [44]. *Desulfovibrio* is a phylotype related to hydrogen-scavenging and sulfate reduction [45] and has been implicated in a variety of diseases, but its relationship with IBD remains controversial, and related research results are conflicting [46]. However, studies have shown that *Desulfovibrio* may be effective in attenuating liver damage by producing acetic acid and regulating hepatic lipid metabolism in mice [47]. The CAPE-emulsion treatment can significantly reduce the abundance of pathogenic bacteria and restore the abundance of probiotic bacteria in DSS-induced colitis mice, suggesting the positive effect of the long-term intake of the CAPE-emulsion on maintaining gut microbiota balance.

In this study, DSS inhibited steroid hormone biosynthesis, steroid biosynthesis, primary bile acid biosynthesis, sphingolipid metabolism, and taurine and hypotaurine metabolism. Among them, the steroid hormone biosynthesis, steroid biosynthesis, and primary bile acid biosynthesis are related to cholesterol metabolism. The inhibition of these three metabolic pathways impedes cholesterol metabolism. Excess cholesterol can lead to an excessive accumulation of fat in the liver, leading to liver inflammation and colorectal cancer [48]. Primary bile acids are endogenous ligands of several transcription factors of the nuclear hormone receptor family, and can directly cause the activation of G protein-coupled bile acid receptor 1, thereby indirectly activating NF-κB and ultimately inducing colorectal cancer [49]. The taurine and hypotaurine metabolic pathways can affect the production of secondary bile acids through enzymatic hydrolysis processes [50]. Studies have also proven that sphingolipid metabolism disorders and their interaction with the gut microbiome are another key to the pathogenesis of IBD [51]. The intake of the CAPE-emulsion can restore the normal function of these metabolic pathways to a certain extent. Some studies have also shown that polyphenols can enhance cholesterol metabolism [52], which is similar to the results of this study.

Spearman correlation analysis confirmed the correlation between gut microbiota imbalance and metabolic disorders in DSS-induced colitis mice. An increase in the abundance of these pathogenic bacteria is accompanied by the suppression of lipid metabolism pathways, which increases the risk of fat accumulation in the liver. Secondary liver injury is one of the most common complications in IBD patients [53]. Therefore, the impact of the CAPE-emulsion on its related gut microbiota and metabolic pathways also implies that the CAPE-emulsion may also have a relieving effect on secondary liver injury complicated by IBD.

## 5. Conclusions

In summary, the colorectal-targeted release of S/O/W emulsion with CAPE can effectively alleviate the symptoms of DSS-induced colitis (such as DAI, colon length, and the pathology of colon sections). Its mechanism includes regulating the expression of inflammation-related cytokines and inhibiting the expression of the P65 subunit in the NF-κB pathway to maintain immune homeostasis. The CAPE-emulsion also promoted the production of the SCFAs and maintained gut microbiota balance by reducing the abundance of pathogenic bacteria and restoring the abundance of probiotics in DSS-induced colitis mice. Fecal metabolomic analysis further confirmed that the CAPE-emulsion can alleviate colitis and its complications by affecting metabolic pathways related to inflammation and cholesterol metabolism. In addition, correlation analysis also demonstrated the interaction between gut microbiota and fecal metabolome in DSS-induced colitis mice.

## Figures and Tables

**Figure 1 nutrients-16-01145-f001:**
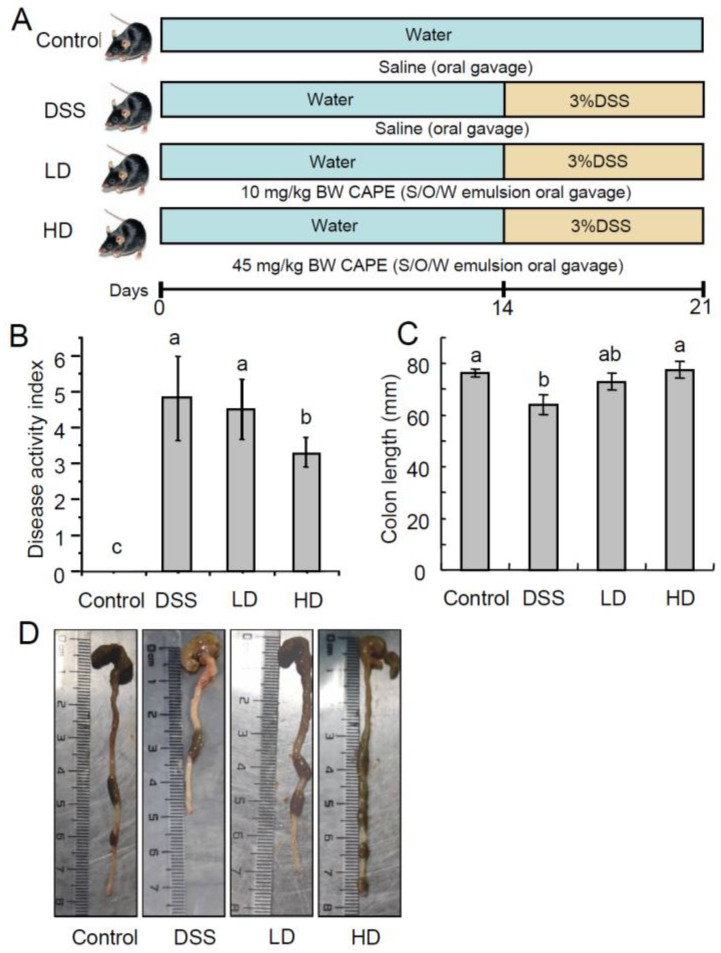
The CAPE-emulsion mitigated symptoms of DSS-induced colitis mice. (**A**) Experimental design, (**B**) DAI, (**C**) colon length, (**D**) representative images of mouse colons. The data are presented as means ± SEM. Different letters represent significant differences among groups *(p* < 0.05, *n* = 10).

**Figure 2 nutrients-16-01145-f002:**
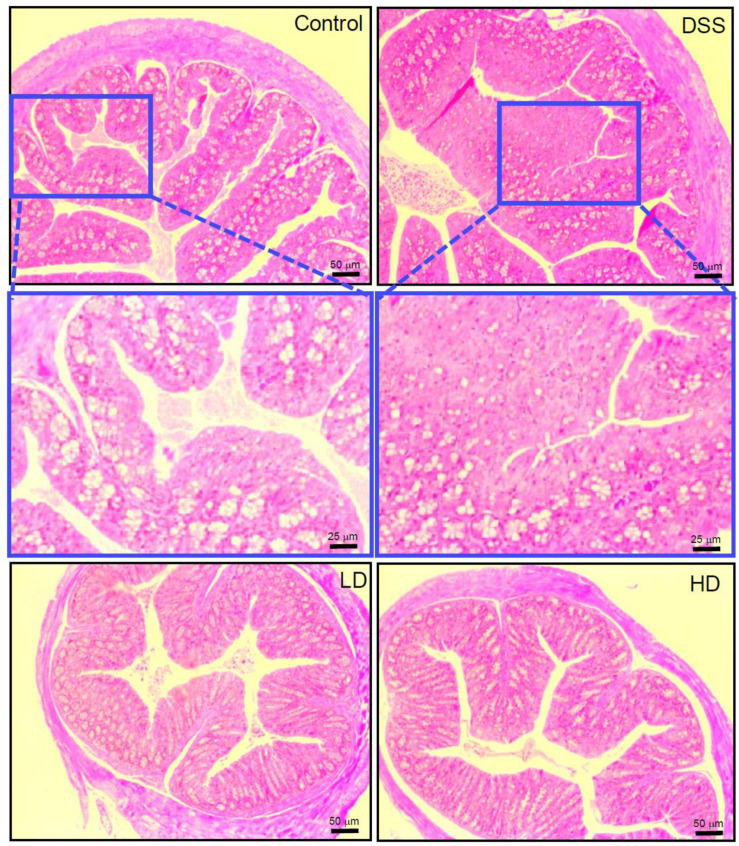
Representative H&E staining images of the colon tissues in mice (100× & 400×).

**Figure 3 nutrients-16-01145-f003:**
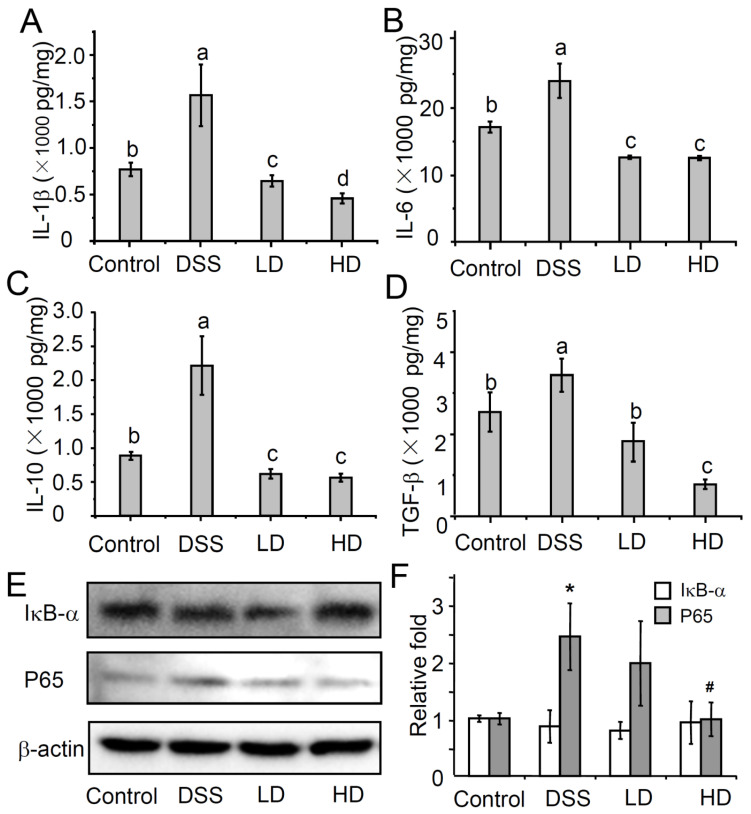
(**A**–**D**) Effects of the CAPE-emulsion on inflammatory cytokines secreted in colonic tissues. Data are expressed as means ± SEM. Different letters represent significant differences among groups (*p* < 0.05, *n* = 4); (**E**) Western blotting for IκB-α and P65 in colonic tissues; (**F**) the densitometry based on Western blotting band of IκB-α or P65 relative to β-actin. Data are expressed as means ± SEM, “*” represent significant differences, *p* < 0.05 vs. Control; “**#**” represent significant differences, *p* < 0.05 vs. DSS (*n* = 3).

**Figure 4 nutrients-16-01145-f004:**
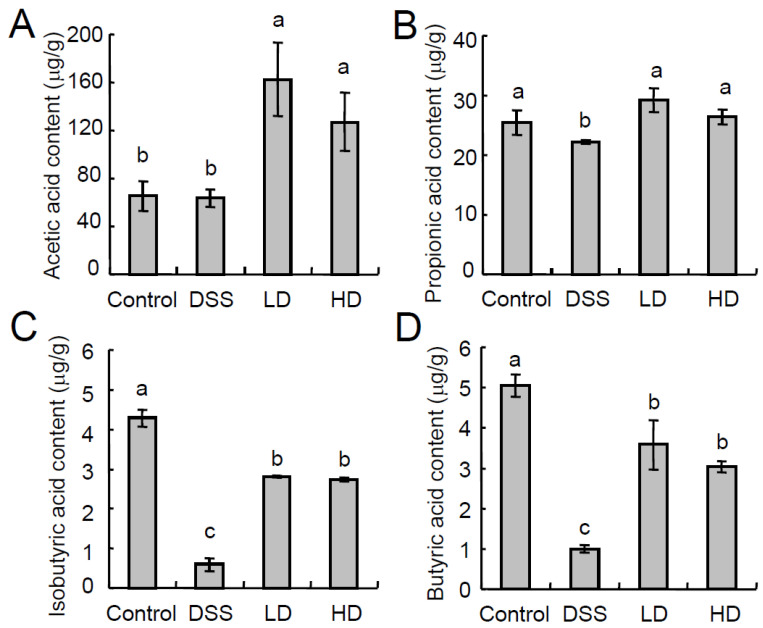
Effects of the CAPE-emulsion on the short-chain fatty acid levels in feces. (**A**) Acetic acid content in feces; (**B**) Propionic acid content in feces; (**C**) Isobutyric acid content in feces; (**D**) Butyric acid content in feces.Data are expressed as means ± SEM. Different letters represent significant differences among groups (*p* < 0.05, *n* = 4).

**Figure 5 nutrients-16-01145-f005:**
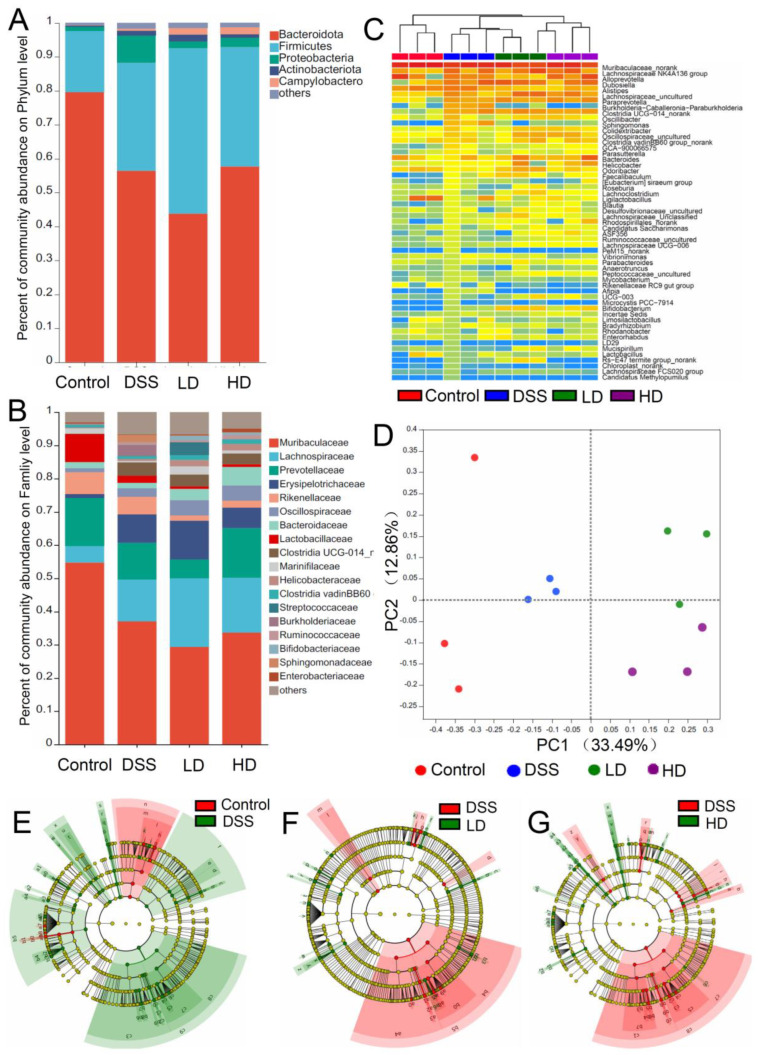
Effects of the CAPE-emulsion on the gut microbiota in the DSS-induced colitis mice. The relative abundances of the gut microbiota at the phylum level (**A**) and the family level (**B**) in each group. (**C**) Cluster analysis and heatmap of the gut microbiota at the genus level based on the relative abundances. (**D**) Two-dimensional PCoA plots of the OUT data. (**E**–**G**) Taxonomic cladogram of the LEfSe analysis from phylum to genus.

**Figure 6 nutrients-16-01145-f006:**
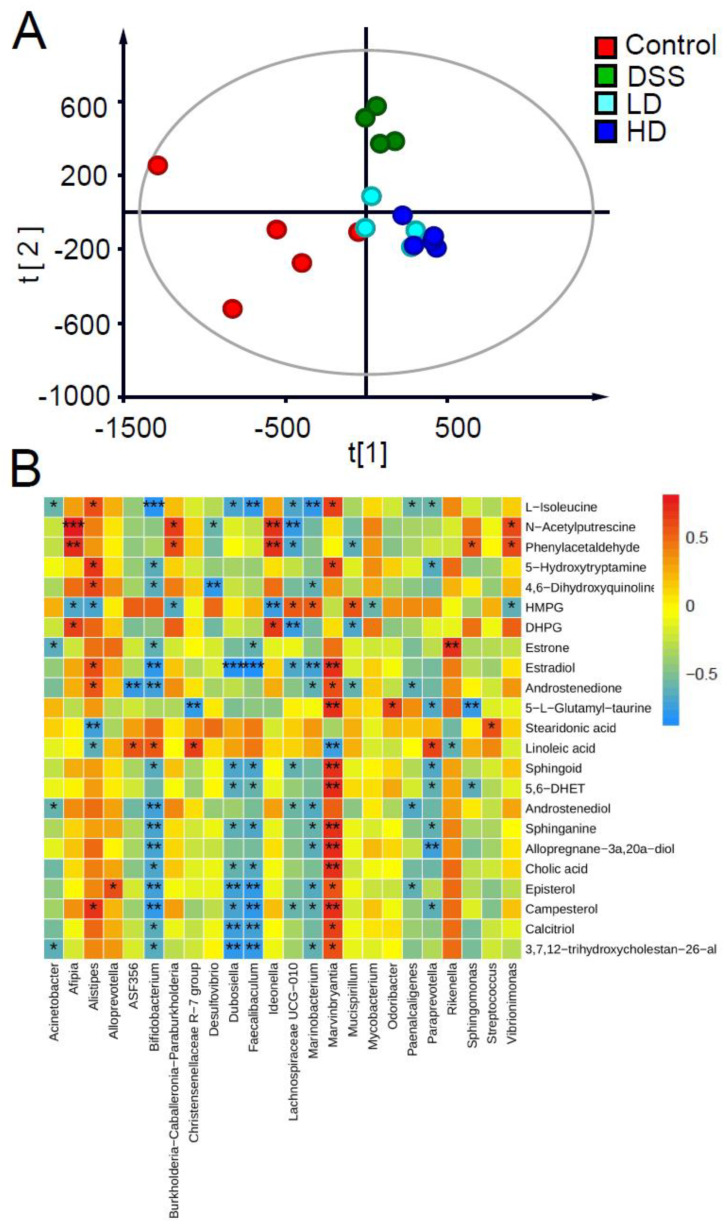
(**A**) PLS-DA score plot of fecal metabolites determined via UPLC-Q-TOF in positive mode (*n* = 4–5). (**B**) Spearman correlation heatmap of key fecal metabolites and biomarkers in gut microbiota. “*” represents a significant correlation, *p* < 0.05; “**” represents a significant correlation, *p* < 0.01; “***” represents a significant correlation, *p* < 0.001.

**Table 1 nutrients-16-01145-t001:** Effects of the CAPE-nanoemulsion on the diversity index of gut microbiota.

	Sobs	Ace	Shannon	Simpson	Chao 1
Control	828.667 ± 29.036 ^a^	994.972 ± 37.129 ^a^	4.344 ± 0.108 ^a^	0.053 ± 0.011 ^a^	1005.394 ± 36.64 ^a^
DSS	546 ± 6.807 ^d^	660.252 ± 43.466 ^b^	4.235 ± 0.112 ^a^	0.05 ± 0.005 ^a^	669.807 ± 53.923 ^c^
LD	653 ± 21.362 ^c^	882.753 ± 85.217 ^a^	4.617 ± 0.222 ^a^	0.031 ± 0.01 ^a^	779.725 ± 49.722 ^bc^
HD	748 ± 17.898 ^b^	834.798 ± 45.16 ^ab^	4.452 ± 0.171 ^a^	0.036 ± 0.008 ^a^	847.809 ± 39.88 ^b^

^a,b,c,d^ Mean ± SEM with different letter in the same column show statistically significantly different (*p* < 0.05, *n* = 3).

**Table 2 nutrients-16-01145-t002:** Biomarkers in fecal metabolites based on UPLC-Q-TOF analysis.

*m*/*z*	Excat Mass (Da)	Elemental Composition	Selected Ion	Postulated Identity	KEGG ID	VIP Values	DSS vs. Control	LD vs. DSS	HD vs. DSS
104.1062	131.0946	C_6_H_13_NO_2_	[M-CO+H]+	L-Isoleucine	C00407	1.9553	—	↓	↓
114.0904	130.1106	C_6_H_14_N_2_O	[M-NH_3_+H]+	N-Acetylputrescine	C02714	3.1699	↑	↓	↓
199.1473	270.162	C_18_H_22_O_2_	[M-C_3_H_4_O_2_+H]+	Estrone	C00468	1.3757	↓	—	↑
201.1631	272.1776	C_18_H_24_O_2_	[M-C_3_H_4_O_2_+H]+	Estradiol	C00951	1.5846	↓	—	—
241.1955	286.1933	C_19_H_26_O_2_	[M-HCOOH+H]+	Androstenedione	C00280	1.3566	—	—	↓
313.2163	290.2246	C_19_H_30_O_2_	[M+Na]+	Androstenediol	C04295	1.1001	↓	↓	↓
339.2877	320.2715	C_21_H_36_O_2_	[M-H_2_O+H]+	Allopregnane-3α,20α-diol	C18042	2.2882	↓	—	↓
421.3475	398.3549	C_28_H_46_O	[M+Na]+	Episterol	C15777	1.1393	↓	↓	↑
423.3622	400.3705	C_28_H_48_O	[M+Na]+	Campesterol	C01789	1.5486	↓	↓	↑
439.3199	416.329	C_27_H_44_O_3_	[M+Na]+	Calcitriol	C01673	1.2781	↓	—	↑
282.2787	299.2824	C_18_H_37_NO_2_	[M-H_2_O+H]+	Sphingoid	C00319	1.4562	↓	—	↑
324.2879	301.2981	C_18_H_39_NO_2_	[M+Na]+	Sphinganine	C00836	1.497	↓	—	↑
281.247	280.2402	C_18_H_32_O_2_	[M+H]+	Linoleic acid	C01595	1.7064	↑	—	—
255.065	254.0573	C_7_H_14_N_2_O_6_S	[M+H]+	5-L-Glutamyl-taurine	C05844	1.2138	↓	↑	↑
185.0800	184.0736	C_9_H_12_O_4_	[M+H]+	HMPG	C05594	2.1462	↓	↑	↑
193.0491	170.0579	C_8_H_10_O_4_	[M+Na]+	DHPG	C05576	1.6714	↑	↓	↓
149.1067	176.095	C_10_H_12_N_2_O	[M-CO+H]+	5-Hydroxytryptamine	C00780	1.1039	↓	—	↓
116.0486	161.0477	C_9_H_7_NO_2_	[M-HCOOH+H]+	4,6-Dihydroxyquinoline	C05639	1.5739	—	↓	↓
121.0644	120.0575	C_8_H_8_O	[M+H]+	Phenylacetaldehyde	C00601	1.2324	↑	↓	↓
277.2176	276.2089	C_18_H_28_O_2_	[M+H]+	Stearidonic acid	C16300	1.1778	—	↑	↑
311.2593	338.2457	C_20_H_34_O_4_	[M-CO+H]+	5,6-DHET	C14772	1.2515	↓	↑	—
391.2843	408.2876	C_24_H_40_O_5_	[M-H_2_O+H]+	Cholic acid	C00695	1.8347	↓	—	↑
457.3308	434.3396	C_6_H_13_NO_2_	[M+Na]+	3,7,12-trihydroxycholestan-26-al	C01301	1.7466	↓	—	↑

↑ indicates significantly increased at *p* < 0.05; ↓ indicates significantly decreased at *p* < 0.05; — indicates no statistical significance (*n* = 4–5).

## Data Availability

All data utilized in the present work can be obtained from the corresponding author on request.

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
