# Peer review of "S/O/W Emulsion with CAPE Ameliorates DSS-Induced Colitis by Regulating NF-κB Pathway, Gut Microbiota and Fecal Metabolome in C57BL/6 Mice"

_nutrients, 2024, doi:10.3390/nu16081145_

Round 1

Reviewer 1 Report

Comments and Suggestions for Authors

The paper nutrients-2939537 presents original research on caffeic acid phenethyl ester (CAPE) as a potential compound to treat inflammatory bowel diseases (IBD) in a DSS-induced colitis mouse model. The authors investigate CAPE effects in terms of disease activity index (DAI), colon length and histopathology, inflammation and oxidative stress, gut microbiota, and faecal metabolome.

I believe the manuscript suffers some major methodological limitations, as follows:

The authors do not mention data related to the CAPE emulsion bioavailability (in PMID 30170869, for instance, CAPE is also used intraperitoneally in a very similar model). Previous research of the same authors (PMID: 36192871) should be better introduced also to justify the two dosages used here.

Paragraph 2.6 and results thereof. I do not understand why the authors refer to secreted cytokines when they analyse a tissue homogenate. Analyses on blood samples are needed to confirm secretion, which are lacking here.

Paragraph 2.7 and results thereof. Western blot against the phosphorylated forms of the tested proteins is needed in order to calculate the ratio phosphoprotein/total protein, i.e. phospho-p65/p65 and phopspo-IκB-α/IκB-α. Conversely, the authors assessed total protein content only, which is not sufficient for the analysis of the activation of the NF-kB pathway.

Additional minor comments below.

Lines 194-195. Results for LD in Figure 1 C and D are not statistically significant.

Paragraph 2.7. The concentration of protease inhibitors is missing. Why were SDS-PAGE gels loaded with different amounts of proteins (30 – 50 ug)?

Figure 2: size bars are missing.

Comments on the Quality of English Language

Despite the use of an English editing service, the manuscript still contains some typos and unclear/truncated sentences.

Reviewer 2 Report

Comments and Suggestions for Authors

The manuscript is interesting, novel and fits into the journal's field of study. It is very well written and the figures and tables are well presented, so that it is easy to read and attractive to readers. The main and almost unique deficiency of the manuscript is its introduction. It is extremely short, lacks relevant references and does not even define important concepts such as S/O/W emulsions. In my opinion, the introduction should be redone or expanded. 

Round 2

Reviewer 1 Report

Comments and Suggestions for Authors

I thank the authors for having addressed my concerns in the first round of revision. However, I would suggest that their manuscript undergoes some additional modifications, as follows:

Amend “bolt” in “blot” at line 141

Please state clearly the ug of proteins loaded in each gel instead of using a range (line 148)

Line 249-250: there is not such a Supplementary Table 3 in the revised material, therefore I am not able to evaluate what the authors refer to. In any case, if the authors did not test phosphorylated proteins in their samples, they must significantly mitigate their comments on the NF-kB pathway (lines 251-254 and 388-391)

The original images of blots show several non-specific bands detected concomitantly with p65. This set of experiments needs significant technical improvement.

Author Response

All revisions in manuscript are marked in red.

I thank the authors for having addressed my concerns in the first round of revision. However, I would suggest that their manuscript undergoes some additional modifications, as follows:

Amend “bolt” in “blot” at line 141

Response: Revised accordingly.

Please state clearly the ug of proteins loaded in each gel instead of using a range (line 148)

Response: Revised accordingly.

Line 249-250: there is not such a Supplementary Table 3 in the revised material, therefore I am not able to evaluate what the authors refer to. In any case, if the authors did not test phosphorylated proteins in their samples, they must significantly mitigate their comments on the NF-kB pathway (lines 251-254 and 388-391)

Response: Revised accordingly in lines 25-26, 256-258, 396-398 and 466-468. The supplementary Table 3 is a mistake, its data is Figure 3F. We have mitigated the comments on the NF-κB pathway in this paper. We believe that the CAPE-emulsion treatment down-regulates the expression of the P65 subunit in the NF-κB pathway, rather than inhibits the activation of the NF-κB pathway.

The original images of blots show several non-specific bands detected concomitantly with p65. This set of experiments needs significant technical improvement.

Response: We will improve the experimental technique and replace the antibody with better specificity in future experiments.